# Changes of in-vivo markers of platelet activation during the menstrual cycle in healthy pre-menopausal female individuals
Valeria Raparelli [1], Marzia Miglionico [2] ✉, Francesca Maiorca [1], Laura Napoleone[3], Lombardi Ludovica[1], Annamaria Sabetta[1], Tania D'Amico[1], Giulio Francesco Romiti [1,4], Andrea Lenzi[2], Cristina Nocella[5], Alessandra D'Amico[6,7], Giacomo Frati [6,8], Sebastiano Sciarretta[6,8], Roberto Carnevale[6,8], Roberto Cangemi [1], Stefania Basili[1,10] & Lucia Stefanini [1,9,10]

## Abstract

**Background** Sex differences in cardiovascular diseases (CVD) are a matter of immediate concern, with platelets playing a pivotal role as major contributors to CVD. While the protective effects of estrogens on vascular health have been largely investigated, the impact of endogenous reproductive hormones on platelet function is less clear. In the SHOW (Sexual Hormones and Hemostasis: Observations for Women Health) study, we aimed to assess the association between the levels of endogenous reproductive hormones and *in-vivo* platelet activation among pre-menopausal healthy female subjects.

**Methods** Serum levels of estradiol, progesterone (PG), testosterone, luteinizing hormone (LH), follicle-stimulating hormone (FSH), and in vivo platelet activation markers were quantified at four time-points across the menstrual cycle (day (d) 1, d5 ± 2, d14 ± 2, and d21 ± 2).

**Results** Among 21 healthy participants (mean age: 30 years), significant variations of thromboxane B2 (TxB2), soluble P-selectin (sP-selectin) and soluble NOX2-derived peptide (sNOX2-dp) are detected across the menstrual cycle. Linear mixed model analysis shows that sP-selection is associated with LH levels ($F = 6.400$, $p = .016$) in a time-dependent manner. TxB2 is associated with FSH across all time-points ($F = 6.051$, $p = .019$) and is significantly reduced on d1 in individuals with self-reported heavy menstrual bleeding. Soluble CD40L (sCD40L) changes most significantly in individuals with an ovulatory cycle (i.e. PG ≥ 3 ng/ml at d21).

**Conclusions** In healthy pre-menopausal female subjects, changes of *in-vivo* platelet activation markers are associated with changes of PG and gonadotropins. These findings suggest the need for caution in the interpretation of platelet biomarkers in clinical studies enrolling pre-menopausal female individuals.

## Plain language summary

Cardiovascular diseases affect men and women differently, with platelets playing a pivotal role in these conditions. In this study, we aimed to evaluate the impact of endogenous reproductive hormones on platelets and vascular health among pre-menopausal healthy females. Hormone levels and platelet activity markers were measured at different phases of the menstrual cycle. The results showed that specific markers of platelet activation varied across the menstrual cycle, in association with changes of gonadotropins and progesterone. Recognition of this sex-specific platelet behavior is essential for the accurate interpretation of clinical studies investigating antiplatelet therapies in premenopausal women, in comparison with male counterparts.

Cardiovascular diseases (CVD) account for 40% of all deaths among European female adults and for substantial morbidity and resource use in the healthcare system[1]. Nonetheless, sex-related differences in the epidemiology and development of CVD are still not completely explained. Among the mechanisms accounting for such differences, the biology of platelets, which exert a pivotal pathogenetic role in atherothrombosis[2], has been investigated over time.

Biological sex is claimed to be a major factor affecting platelet reactivity across the people's lifespan[3], as there are female-specific hemostatic changes

needed for childbirth[4], sex-based specificities of CVD have been documented[5], and RNA-sequencing technology identified approximately 50 messenger RNAs that are differentially expressed in human platelets between sexes[6]. Thus, there is an urgent need to incorporate sex-specific questions when it comes to a better and informative understanding of CV health and disease[7].

In the last decade, both preclinical and clinical studies have provided clues suggesting that a different modulation of platelet function might influence the clinical course of CVD among female individuals, who fare worse than their male counterparts when they experience CV events[8]. The platelets of pre-menopausal female subjects without known CVD are more reactive in response to standard concentrations of agonists and have a higher propensity to aggregate and adhere than those of male individuals[9]. The observed sexual dimorphism could be due to the different platelet transcriptomic signature between males and females[6] and/or to non-genomic effects of sex hormones directly on platelets, since they express the estrogen receptor (ER)[10]. Atheroma-protective effects of both endogenous and exogenous estrogens have been largely reported[11]. Nevertheless, the Women's Health Initiative study did not show a coronary protective effect of estrogen (i.e., conjugated equine estrogens) in postmenopausal female participants, possibly because of the inappropriate timing of the therapy and the type of associated progestin (i.e., medroxyprogesterone acetate)[12]. Studies focused on sex differences in platelet function driven by either endogenous (i.e., across the menstrual cycle) or exogenous estrogens are conflicting[13–17] and limited, as they only investigate whether there were changes in platelet reactivity and adhesiveness upon stimulation ex vivo. However, aggregation at sites of vascular injury is not the sole function of platelets. Indeed, platelets are crucially involved in maintaining vascular integrity at the steady state[18] and at sites of inflammation[19], in regulating the vascular tone[20,21], and modulating immune cell circulation and activation[22]. Thus, we hypothesize that platelets become activated in healthy female subjects without a vascular injury to ensure vascular homeostasis as the hormone levels change during the menstrual cycle. Based on this hypothesis, our study aimed at assessing how in vivo platelet activation markers change with respect to the physiological fluctuations of endogenous sex hormones in pre-menopausal healthy female individuals.

In our cohort of 21 healthy women, platelet activation markers show significant fluctuation across the menstrual cycle. In linear mixed-effects models, thromboxane B2 (TxB2) shows a consistent positive association with follicle-stimulating hormone (FSH) throughout the cycle and a reduction on day 1 among women reporting heavy menstrual bleeding. Luteinizing hormone (LH) is significantly associated with soluble P-selectin (sP-selectin), as supported by a significant interaction between LH and time. Finally, soluble CD40L (sCD40L) increases significantly at day 14 in participants with biochemically confirmed ovulation (progesterone ≥3 ng/mL at day 21).

## Methods

The SHOW (*Sexual Hormones and Hemostasis: Observations for Women Health*) study (Clinicaltrial.gov NCT02055001) was designed as an observational study to investigate the relationship between platelet function and sex hormones during physiological regular menstrual cycles (28–30 days) in healthy pre-menopausal female subjects aged from 18 to 40 years. Participants were recruited at the Department of Experimental Medicine, University Hospital Policlinico Umberto I Rome, Italy in 2014. The following exclusion criteria were applied: i) history of recent (within the last 2 months) use of anticoagulants, antifibrinolytics, non-steroidal anti-inflammatory medications, combined oral contraceptives (COC), and progestogens; ii) pregnancy; iii) presence of kidney, liver, heart, endocrine diseases or infective diseases for at least 2 months prior to the study; and iv) use of chronic medications for any known disease. Written informed consent was obtained from all participants according to Italian regulations, and the experimental procedure was approved by the Institutional Review Board at Sapienza University of Rome (protocol number: 2972/14.11.2013) and was conducted in accordance with the Declaration of Helsinki.

The primary outcome of the study was to assess the changes in circulating levels of biomarkers reflecting the in vivo activation status of platelets during a regular menstrual cycle. Specifically, the time points for blood sampling were day (d) 1 (baseline), d5 ± 2 (early follicular phase/menstruation phase), d14 ± 2 (late follicular phase/expected ovulation), and d21 ± 2 (midluteal phase) of the menstrual cycle.

### Clinical data

Participants' self-reported clinically relevant information regarding demographics, lifestyle behaviors, and risk factors for CVD: being a current smoker, alcohol consumption defined as greater than 1 standard alcoholic unit per day, and being engaged in a leisurely physical activity at least once per week. Physical data, including systolic and diastolic blood pressures, body mass index (BMI), and heart rate (beats per minute) were measured at enrollment.

Reproductive history (i.e., prior pregnancies, previous use of oral contraceptives) and features of the menstrual cycle were collected through an interview with clinicians.

Specifically, participants were asked to report the age at menarche and a qualitative estimation of their usual menstrual bleeding volume, defining heavy menstrual bleeding as large volume interfering with physical, emotional, or social quality of life.

### Circulating biomarkers of platelet activation

Plasma levels of the soluble ectodomains of P-selectin (sP-selectin) and CD40L (sCD40L) were measured using a commercial immunoassay (Supplementary Table 1), and values are expressed as ng/mL. Intra-assay and inter-assay coefficients of variation were 7% and 9% for sCD40L and 5.6% and 7.5% for sP-selectin.

Thromboxane B2 (TxB2), the stable analog of thromboxane A2 (TxA2), was measured with an ELISA commercial kit (Supplementary Table 1). Values were expressed as pg/ml; intra- and inter-assay coefficients of variation for TXB2 were <8% and <10%, respectively.

The soluble NOX2-derived peptide (sNOX2-dp) was measured as by ELISA, as previously described[23] (Supplementary Table 1). Briefly, the peptide is recognized by binding to a specific monoclonal antibody against the amino acid sequence (224–268), the extracellular portion of NOX2. Values were expressed as pg/ml; intra-assay and inter-assay coefficients of variation were <10%. The product numbers of the kits used to measure platelet biomarkers are specified in more detail in Supplementary Table 1.

### Serum sex hormones levels

The concentration of hormones in the participants' sera was measured in batch by the Laboratory of the Department of Experimental Medicine (Section of Medical Pathophysiology), Sapienza University of Rome in serum samples stored at −80 °C.

Serum estradiol (E2), luteinizing hormone (LH), follicle-stimulating hormone (FSH), progesterone (PG), and testosterone (TE) were measured by chemiluminescent micro-particle immunoassay (CMIA, Architect System) (Abbott Laboratories, IL, USA). Levels of PG ≥ 3 ng/ml at d21 were used to define the occurrence of the ovulatory cycle based on previous literature[24–27].

### Statistics and reproducibility

Continuous variables were expressed as mean ± standard deviation (SD) or median [interquartile range (IQR)], and categorical variables as number (*n*) and percentage (%).

Distribution of continuous variables was inspected graphically using quantile-quantile (Q-Q) plots. Group comparisons were performed using the Friedman non-parametric test, with post-hoc Dunn-Bonferroni tests when variables were non-normally distributed, matching the repeated measures from the same individual. Linear mixed-effects models were used to assess the longitudinal associations between hormonal levels and markers

of platelet activation across the menstrual cycle. The models included fixed effects for time (menstrual cycle phases), hormone levels, and their interaction (time × hormone). The time variable was treated as categorical (T1, T2, T3, T4), representing standardized cycle days. Continuous predictors were entered as covariates. Models were estimated using the restricted maximum likelihood approach with an unstructured covariance matrix to model correlations between repeated measures. Significance of fixed effects was tested using Type III F-tests, and a p-value < 0.05 was considered statistically significant.

The principal component analysis (PCA) was conducted using the platelet variables (TxB2, sCD40L, sP-selectin, NOX2dp) at different time points to calculate the principal components (PCs). The criteria for PCs selection were eigenvalues greater than 1. The output PC scores plot was graphed to show different symbol fill colors for different time points. Analyses were performed using computer software packages (SPSS-25.0, SPSS Inc., Chicago, IL; GraphPad Prism, Version 10.2.3). A P value <.05 was considered statistically significant.

### Sample size determination

A minimum sample size of 21 pre-menopausal female subjects was estimated. With a statistical power of 90% and type I error of 5% (two-sided), we aimed to detect a 30% change in thromboxane concentrations between day 1 (estimated level 210 pg/mg creatinine with a SD of 40 pg/mg creatinine based on prior published data[28] and day 5 (early follicular phase) of the menstrual cycle.

### Results

Twenty-one healthy female young adults (mean age ± SD = 30.3 ± 3.1 years) with normal weight (BMI, mean ± SD = 21.2 ± 2.2) were enrolled. Baseline characteristics of the population included are reported in Table 1.

Overall, participants experienced the first menstrual period around 12 years of age and had an average menstrual cycle of 29 ± 3 days. Most participants (82%) reported a mild consumption of alcohol (3 times per week), and around 24% were smokers (less than 8 cigarettes per day). 36.4% of study participants reported a moderate-to-intense physical activity (≥ 3 times per week). Only 2 individuals had a prior pregnancy and reported no complications. Half of the participants (57%) reported prior use of COC. Only one subject had a family history of early CVD (Table 1).

Serum levels of FSH were maximal at baseline and decreased significantly during the menstrual cycle (between d1-21, p = 0.0005), LH peaked at the time of ovulation (between d1-14, p = 0.036; d14-21, p = 0.0008), while PG increased gradually to reach maximal concentrations in the mid-luteal phase (between d1-21, p = 0.05; d5-14, p = 0.0016). E2 and TE concentrations peaked on day 14 (Fig. 1).

When considering the fold change of the hormone levels relative to the first day of the cycle, E2 was the hormone that increased the most in the overall cohort (median [interquartile range] E2 fold change d14 = 2.5[0.95–4.6]) (Fig. S1). However, 52% (11/21) of the study participants had PG levels below 3 ng/ml on d21 (median [IQR]: 0.2 [0.1–0.3]), that is a threshold to identify a non-ovulatory cycle in women with normal-length menstrual cycles[24–27]. Individuals with a non-ovulatory cycle also had significantly lower E2 concentration (p = 0.0071), no detectable LH peak on d14 (p = 0.0003) and significantly lower fold changes of FSH levels on d5 (p = 0.0007) and d14 (p = 0.0002) (i.e., follicular phase). Conversely, participants who had an ovulatory cycle (median PG [IQR] = 11.4 [7.9–18.1]), experienced a 30-fold change of PG levels and a more gradual 5-fold change of the E2 levels which peaked on d21, a significant 2-fold increase of LH that peaked on d14 and FSH levels that increased in the follicular phase and decreased in the luteal phase (Figure S2).

### In vivo markers of platelet activation during the menstrual cycle

To assess platelet activation during the menstrual cycle, we measured the circulating concentration of TxB2, the stable analog of TxA2 (Fig. 2A), sNOX2-dp (Fig. 2B), sP-selectin (Fig. 2C) and sCD40L (Fig. 2D).

TxB2 concentration significantly increased between d1 and d5, i.e. during menstruation (p =0.0075), and between d1 and d21 (p < 0.0001), as well as between d14 and d21 (p < 0.0001) (Fig. 2B). The highest TxB2 concentrations were measured in the midluteal phase (Fig. 2A), when FSH is at its minimum (Fig. 1A).

Based on a linear mixed-effects model that examines the effect of hormones on TxB2 concentrations over time, we found that TxB2 associated significantly with FSH across all time points (F = 6.051, p = 0.019). No significant interaction was found between time and FSH (p = 0.075), suggesting that the relationship between FSH and TxB2 remains relatively stable throughout the menstrual cycle. No significant associations were found between TxB2 and the other hormones tested.

### Table 1 | Baseline characteristics of the participants enrolled in the SHOW study

| Baseline Characteristics | Overall Cohort N = 21 | Non-Ovulatory cycle N = 11 | Ovulatory cycle N = 10 | p-value |
|---|---|---|---|---|
| Age, years | 30.3 ± 3.1 | 31.7 ± 2.9 | 28.8 ± 2.8 | 0.03 |
| Weight, kg | 58.7 ± 8.3 | 58.5 ± 9.7 | 59.3 ± 7.3 | 0.82 |
| Height, m | 1.7 ± 0.1 | 1.7 ± 0.1 | 1.1 ± 0.1 | 0.90 |
| BMI | 21.2 ± 2.2 | 20.8 ± 2.1 | 21.3 ± 2.2 | 0.90 |
| SBP, mmHg | 110.5 ± 7.9 | 108.2 ± 8.4 | 112.0 ± 6.7 | 0.03 |
| DBP, mmHg | 69.3 ± 9.8 | 68.6 ± 11.0 | 69.0 ± 8.7 | 0.94 |
| HR, beats/min | 68.4 ± 7.0 | 67.5 ± 6.8 | 69.7 ± 7.6 | 0.48 |
| Menstrual Cycle Onset, years | 12.2 ± 1.2 | 12.3 ± 1.3 | 12.3 ± 1.2 | 0.94 |
| Duration of menstrual cycle, days | 29.4 ± 3.1 | 28.5 ± 2.8 | 30.6 ± 3.3 | 0.12 |
| Self-reported heavy bleeding during menstrual cycle, n (%) | 5 (24) | 1 (9.1) | 4 (40.0) | 0.15 |
| Alcohol mild consumption, n (%) | 18 (82) | 9 (81.8) | 8 (72.7) | 0.99 |
| Smoking n (%) | 5 (24) | 2 (18.2) | 3 (27.3) | 0.63 |
| Prior pregnancy n (%) | 2 (9) | 2 (18.2) | 0 (0.0) | 0.48 |
| Prior COC use n (%) | 12 (54) | 8 (72.7) | 4 (40.0) | 0.20 |
| Family history of early CVD n (%) | 1 (4) | 1 (9.1) | 0 (0.0) | 0.99 |

The data is shown for the overall cohort and stratified based on the occurrence of the ovulatory cycle. Participants with PG ≥ 3 ng/ml were defined as subjects with an ovulatory cycle. Data are expressed as mean ± SD unless otherwise specified. *BMI* body mass index, *SBP* systolic blood pressure, *DBP* diastolic blood pressure, *HR* Heart Rate, *COC* combined oral contraceptives, *CVD* cardiovascular diseases. p-values were calculated by a two-sided Mann-Witney test.

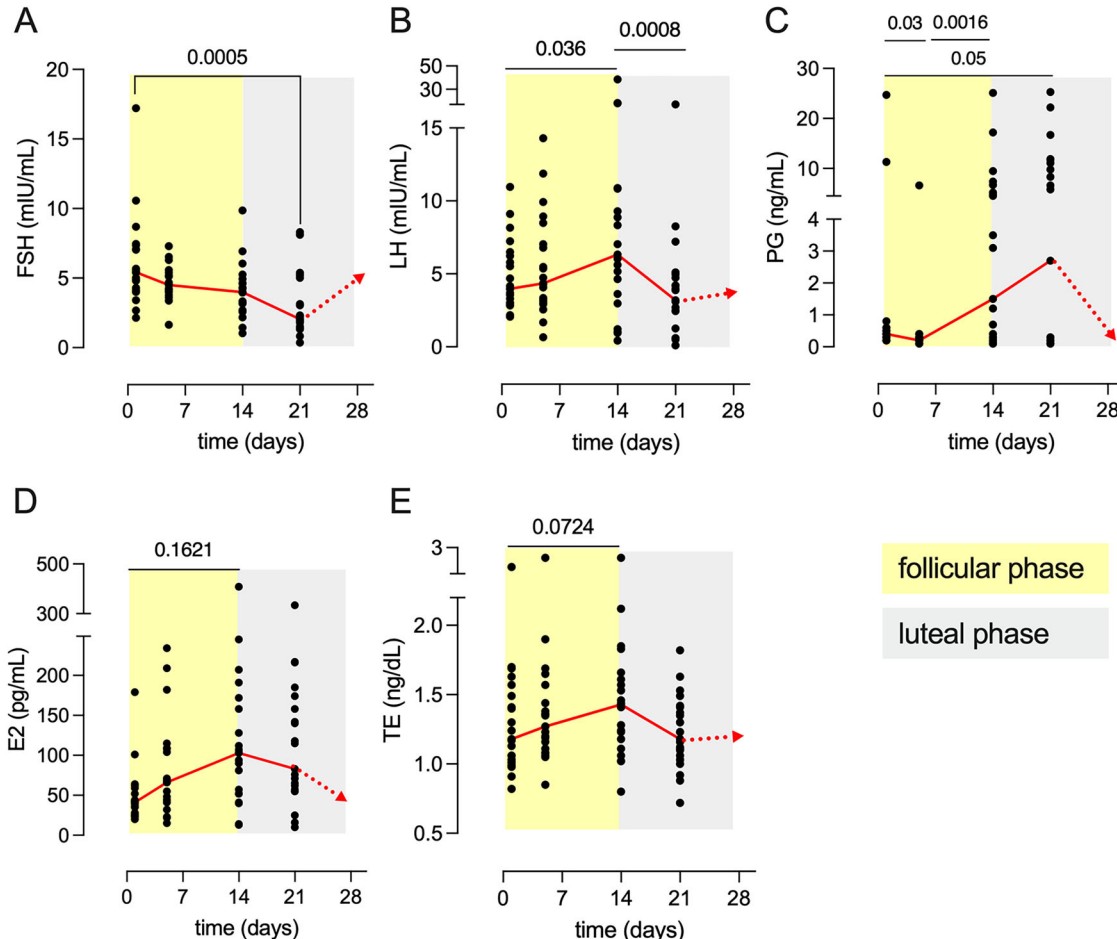

**Fig. 1 | Serum sex hormone levels during the menstrual cycle.** Concentrations of **A** follicle-stimulating hormone (FSH), **B** luteinizing hormone (LH), **C** Progesterone (PG), **D** estradiol (E2), and **E** testosterone (TE) for each study participant ($n = 21$) are shown at each time point. The follicular and the luteal phase are highlighted with yellow and grey background, respectively. The red continuous line shows the median for each time point. The red dotted line between day 21 and the end of the cycle is the predicted concentration assuming a regular cycle will follow. Friedman's two-sided test for repeated measures and Dunn's multiple comparison test were conducted to assess statistical significance.

sNOX2-dp concentration significantly increased ($p = 0.043$) between d1 and d5 (i.e., during menstruation) but decreased ($p = 0.043$) by d14 (in the late follicular/ovulation phase) (Fig. 2B). However, linear mixed models showed no significant interactions between sNOX2-dp and the levels of hormones measured.

sP-selectin levels significantly increased during menstruation ($p = 0.043$) and in the luteal phase ($p = 0.025$) (Fig. 2C).

A linear mixed-effects model indicated that LH levels associated significantly with sP-Selectin (F = 6.400, $p = 0.016$), and this association varied across the menstrual cycle phases as supported by a significant interaction between LH and time (F = 4.200, $p = 0.013$). The sCD40L concentration appeared to increase by d5 but did not reach statistical significance due to high variance (Fig. 2D). On further analysis we observed that on day 5 the study participants clustered in two distinct populations with significantly different levels of plasmatic sCD40L (Fig. S3). The subjects with high sCD40L at the end of menstruation ($n = 13/21$) were among the participants with the highest levels of serum PG and experienced the peak of E2 concentration during the mid-luteal phase rather than on day 14. A linear mixed model showed a borderline association with PG levels (F = 4.308, $p = 0.051$) and a significant interaction between PG and time (F = 4.289, $p = 0.013$) suggesting that the association between PG and sCD40L depends on the menstrual phase. Consistently, when we stratified the fold change of the platelet activation markers based on the PG levels (Fig. 3), we observed that study participants with an ovulatory cycle (PG ≥ 3 ng/ml) had 3-fold higher levels of sCD40L by d14 ($p = 0.03$) but displayed no changes in the sP-

selectin levels over time, while participants with a non-ovulatory cycle (PG < 3 ng/ml) experienced significant changes of the sP-selectin levels over time ($p = .0098$ on d21), but did not experience changes in sCD40L plasmatic concentration.

In sum, we detected significant changes of all in vivo markers of platelet activation across the menstrual cycle, except for sCD40L that changed significantly only in individuals who had an ovulatory cycle.

## Principal Component Analysis of in vivo markers of platelet activation during the menstrual cycle

Since we detected changes in the levels of in vivo markers of platelet activation during the menstrual cycle we performed a principal component analysis (PCA) to assess in an unsupervised manner if combinations of these variables can discriminate the different stages of the menstrual cycle. Combinations of the 4 platelet-related variables generated 4 principal components (PC) and we retained the first two that had a cumulative proportion of variance of 60% and the largest eigenvalues. PC1 negatively associated with TxB2 (inducer of vasoconstriction and platelet activation) and sNOX2-dp (marker of oxidative stress) and positively associated with sCD40L and sP-selectin (both markers of platelet degranulation and platelet-leukocyte crosstalk). PC2 had a large positive correlation with TxB2, sP-selectin and sCD40L (platelet activation markers) and less with sNOX2-dp that is, among all the measured biomarkers, the one that is less platelet specific (Fig. 4A).

**Fig. 2 | In vivo biomarkers of platelet activation during the menstrual cycle.** Concentrations of **A** thromboxane B2 (TxB2), **B** soluble NOX2-derived peptide (sNOX2-dp), **C** soluble P-selectin (sP-selectin) and **D** soluble CD40 ligand (sCD40L), for each study participant ($n = 21$) are shown at each time point. The follicular and the luteal phase are highlighted with a yellow and grey background, respectively. The red continuous line shows the median for each time point. The red dotted line between day 21 and the end of the cycle is the predicted concentration assuming a regular cycle will follow. Friedman two-sided test for repeated measures and Dunn's multiple comparison test were conducted to assess statistical significance.

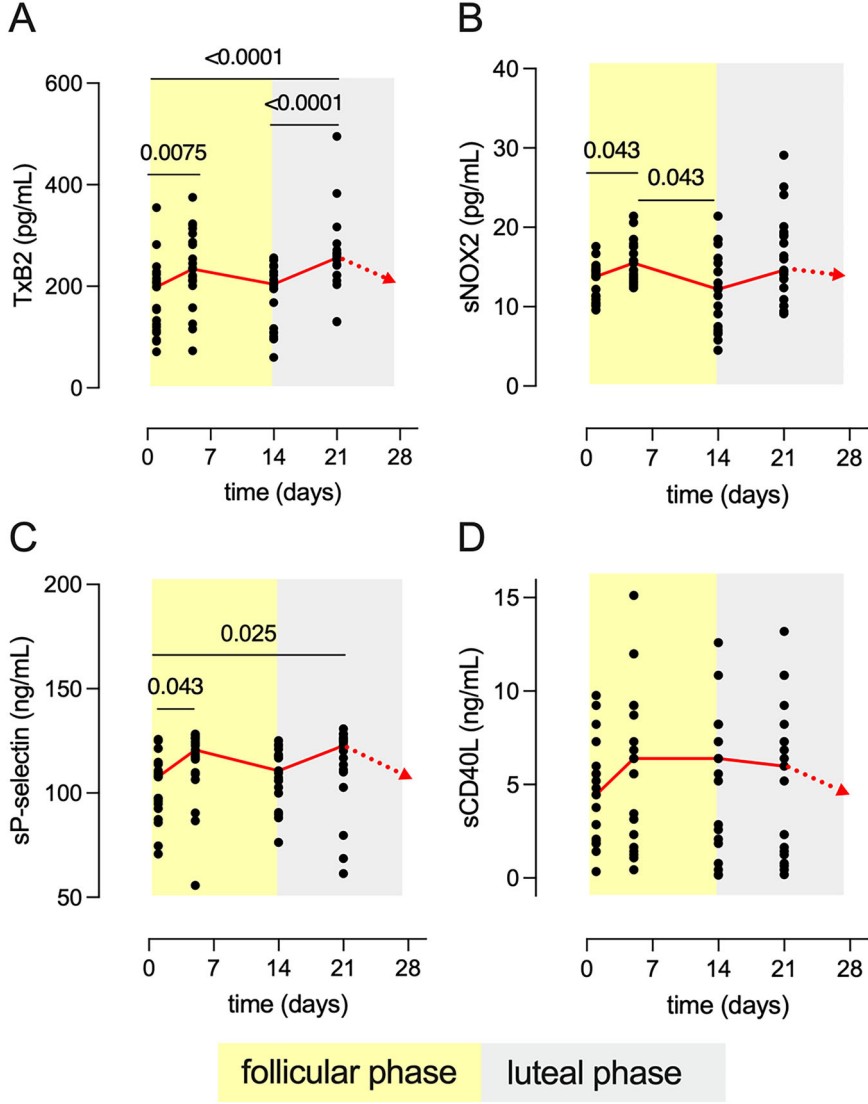

follicular phase    luteal phase

On the PC1/PC2 score plot, the participants formed distinct clusters based on the time-point. At the start of menstruation (day 1), the PC2 scores were negative, i.e., platelet activation markers were low, while after menstruation (day 5), the PC2 scores were positive, reflecting that platelet activation had occurred during menstruation. On day 14, the PC2 scores shifted downward, indicating an attenuation of platelet activation at the time of ovulation. On day 21 the PC2 scores increased again indicating that platelet activation occurred also in the luteal phase, but the cluster shifted to the left, suggesting that different platelet pathways are activated during menstruation and in the luteal phase (Fig. 4B). In sum, this unbiased approach confirms that platelet activation markers change not only at the end of menstruation/early follicular phase, when platelets are required for hemostasis, but also in the mid-luteal phase.

**Platelet in vivo markers that associate with heavy menstrual bleeding**

To identify which features are more relevant to control hemostasis during menstruation we stratified the study participants based on the self-reported volume of menstrual bleeding. Notably, heavy menstrual bleeding was associated with significantly lower levels of TxB2 at baseline (d5) (Fig. 5A), even when considering only the participants with ovulatory cycle (PG > 3 ng/ml) (Fig. S4). Moreover, heavy bleeders had significantly higher levels of sNOX2-dp on d14 (Fig. 5B), but this was not the case when considering only the participants with ovulatory cycle (Fig. S4). No other differences were detected between the levels of platelet in vivo biomarkers at all time-points (Fig. 5C, D). Thus, low TxB2 concentration at the beginning of menstruation is the main hemostatic feature associated with self-reported heavy menstrual bleeding.

## Discussion

In this study we provide evidence that blood biomarkers of in vivo platelet activation change during the menstrual cycle of healthy pre-menopausal female individuals without any overt trauma or inflammatory condition.

The main knowledge increments compared with prior works are that: (i) we detect a significant relationship between the levels of *in-vivo* platelet activation markers and hormones that change during the menstrual cycle; (ii) we identify blood biomarkers that are differentially modulated in female subjects experiencing ovulatory and non-ovulatory cycles (Fig. 6); and (iii) we observe that the main hemostatic feature associated with self-reported heavy menstrual bleeding is a lower TxB2 concentration before menstruation (Fig. 5).

Previous works using ex vivo platelet stimulation in standard aggregometry to establish a relationship between the phases of the menstrual cycle and platelet function is conflicting. Several studies have reported greater agonist-induced platelet aggregation in younger women in reproductive age as compared with age-matched men[14,16,17] identifying in testosterone level the major driver for male lower platelet aggregation[16]. Melamed et al. observed an agonist-dependent association between the phases of the

**Fig. 3 | In vivo biomarkers of platelet activation in individuals with ovulatory and non-ovulatory menstrual cycles.** Mean concentrations of **A** thromboxane B2 (TxB2), **B** soluble NOX2-derived peptide (sNOX2-dp), **C** soluble CD40 ligand (sCD40L, $p$ = *0.03) and **D** soluble P-selectin (sP-selectin, $p$ = **0.0098), in individuals with PG ≥ 3 ng/ml (ovulatory cycle, light blue, $n$ = 10) and in individuals with PG < 3 ng/ml (non-ovulatory cycle, green, $n$ = 11). Two-way ANOVA with Sydak's multiple comparisons test were conducted to assess statistical significance.

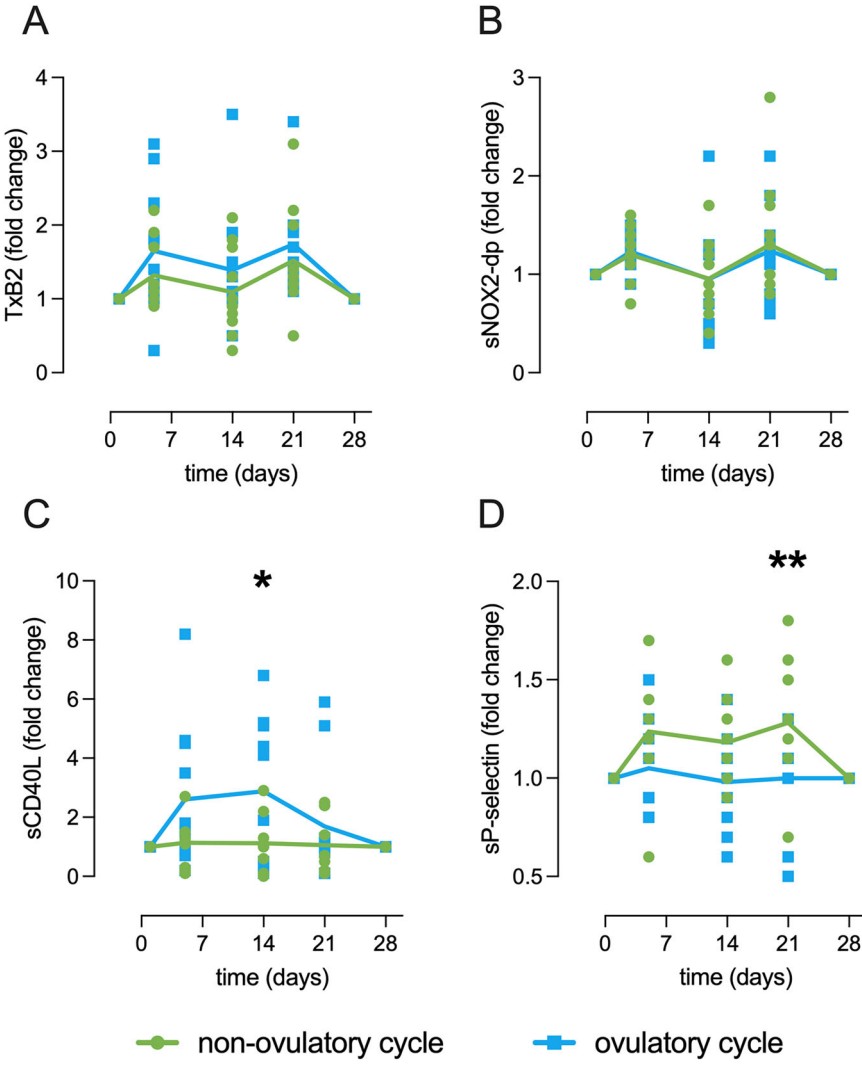

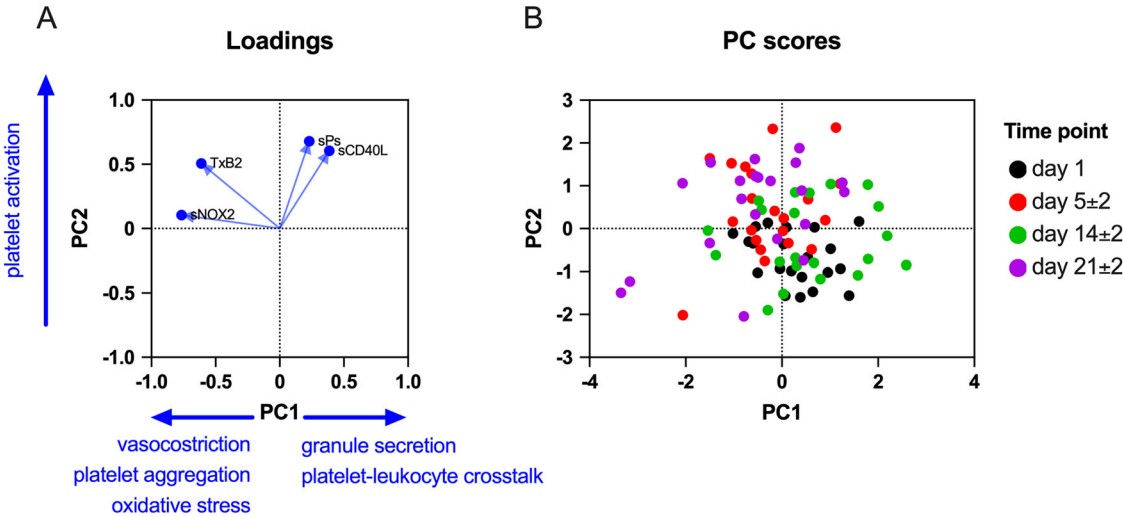

**Fig. 4 | Principal component analysis of platelet activation markers during the menstrual cycle. A** Graph showing the loading and the direction of association of each input variable to the retained principal component. **B** Score plot showing clusters of samples in the different phases of the menstrual cycle, which are graphed with different symbol color fillings.

**Fig. 5 | Platelet activation biomarkers that associate with heavy menstrual bleeding.** Bar graphs of the **A** thromboxane B2 (TxB2) (normal vs heavy at d1, *$p$ = 0.05; d1 vs d21, **$p$ = 0.006; d14 vs d21, **$p$ = 0.006), (**B**) soluble NOX2-derived peptide (sNOX2-dp) (normal vs heavy at d14, *$p$ = 0.043; d5 vs d14, **$p$ = 0.005; d14 vs d21, **$p$ = 0.002), **C** soluble P-selectin (sP-selectin) and **D** soluble CD40 ligand (sCD40L) levels stratified based on self-reported heavy (grey with circular dots, $n$ = 5) or normal (white with square dots, $n$ = 16) bleeding during menstruation. Data were shown as median ± interquartile range (IQR). Ordinary two-way ANOVA test with Šidák multiple comparisons test was used for intergroup and intragroup analysis. *$p$ < 0.05; **$p$ < 0.01; ***$p$ < 0.001; ****$p$ < 0.0001.

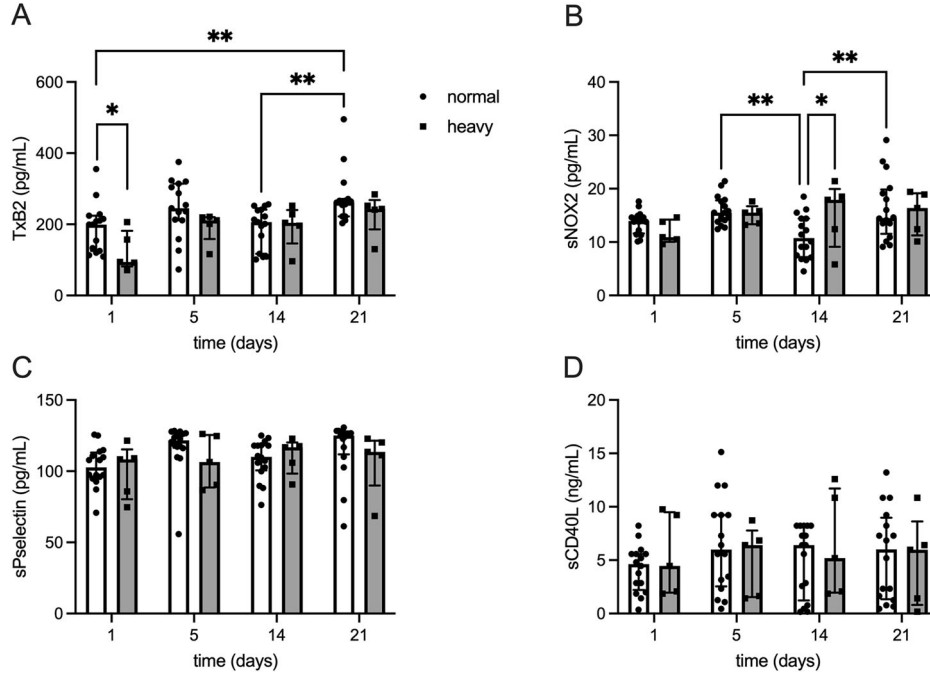

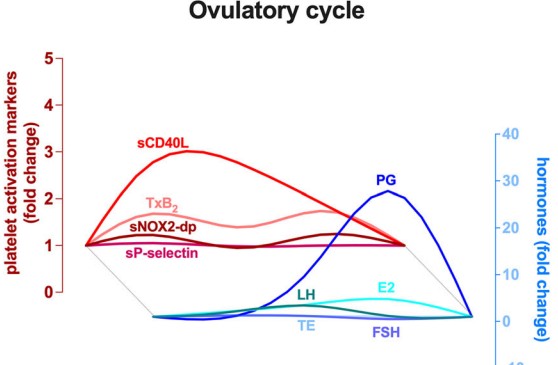

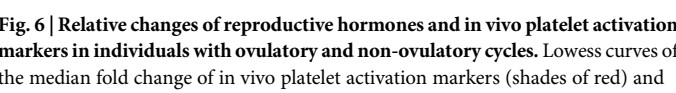

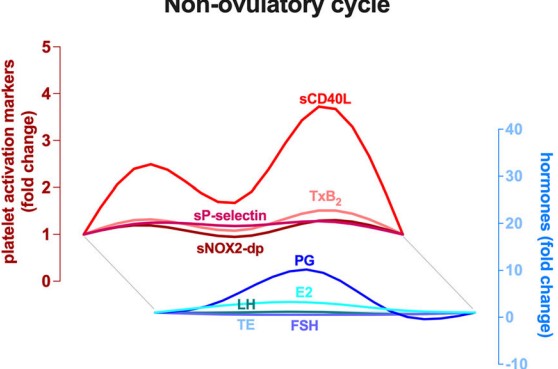

**Fig. 6 | Relative changes of reproductive hormones and in vivo platelet activation markers in individuals with ovulatory and non-ovulatory cycles.** Lowess curves of the median fold change of in vivo platelet activation markers (shades of red) and reproductive hormones (shades of blue) over the course of one menstrual cycle in individuals with PG ≥ 3 ng/ml (ovulatory cycle, $n$ = 10) and in individuals with PG < 3 ng/ml (non-ovulatory cycle, $n$ = 11).

menstrual cycle and platelet reactivity, but platelet aggregation was consistently lower in the mid-luteal phase irrespective of the agonist used[14]. Conversely, another LTA-based study showed that platelets were less prone to aggregation in follicular phase than in mid-luteal phase, depending on the agonist used[29] while Berlin G et al, in accordance with Yee et al. reported that the agonist-induced aggregation response was not significantly influenced by sex hormones variations along the menstrual cycle[30,31]. These inconsistent results may be due to differences in experimental procedures and to the masking effect of ex vivo agonist stimulation. In our study, we did not examine platelet reactivity *ex vivo*, but we measured stable blood biomarkers that reflect in vivo platelet activation. Specifically, we monitored the levels of four established biomarkers that reflect different aspects of platelet activation (Supplementary Table 2). This has led to the interesting observations that platelets are not only activated during menstruation, when their hemostatic function is required, but also in the mid-luteal phase. Moreover, we found that different platelet biomarkers change with different patterns over time and are associated with changes in different hormones. In particular, we find that while sCD40L is the platelet biomarker that increases the most during the follicular phase of ovulatory cycles, low TxB2 at the beginning of the menstrual cycle is the most distinctive feature of female

subjects who had self-reported heavy bleeding during menstruation. These results suggest that platelets have multiple roles in regulating female vascular homeostasis, that go beyond classical hemostasis. Based on a linear mixed model, we detected a significant association between TxB2 and FSH, reflecting that TxA2 increases throughout the cycle (reaching maximal levels in the luteal phase) as FSH levels decline. We also detected a time-dependent association between the levels of sP-selectin and the concentration of LH, as we observed sP-selectin levels decline in the late follicular phase, when LH grows sharply. Since there is no evidence that platelets express the FSH receptor and the luteinizing hormone/choriogonadotropin (LHCG) receptor, the observed associations between FSH and LH with the platelet activation markers are likely indirect. There is growing evidence that gonadotropins modulate endothelial cells, for instance by promoting the production of nitric oxide (NO)[32] a well-established platelet inhibitor. Thus, the inverse relationship between the levels of FSH and LH with TxB2 and sP-selectin could be at least in part due to the regulation of the circulating NO levels.

We also detect an inverse association between sCD40L and PG levels (Fig. 6), particularly evident among individuals who had an ovulatory cycle. In these subjects, sCD40L increases by 3-fold during the follicular phase,

while in subjects with a non-ovulatory cycle, sCD40L reaches its peak in the mid-luteal phase.

Previous studies focused mainly on the effect of E2 on platelet function since platelets express the estrogen receptor[10], while the progesterone receptor (PGR, NR3C3) has not been detected in transcriptomics and proteomics studies of platelets. However, the serum levels and the cellular response evoked by PG and E2 are tightly linked, and early studies have documented a direct non-genomic effect of PG[33] on platelet activation *in vitro*. Since we detect a relationship between sCD40L and PG, but not with E2, our study suggests that research should not only focus on E2 to interpret the relationship between the reproductive system and female health in general[24–27]. Although the statistical associations shown are far from proving any causality and will need to be confirmed in larger cohorts, changes in sCD40L levels in ovulatory/non-ovulatory cycles may reflect the different requirements of platelets in these two distinct scenarios. Further studies may lead to a deeper understanding of the physiological role of platelets in maintaining homeostasis during the menstrual cycle.

In comparison to prior works, the strengths of our study are that: (a) we relate all the reproductive hormones (not only E2 and TE) to platelet function across 4 time points; (b) we employ reproducible assays that are easily transferrable; and (c) we do not study platelet reactivity to agonists but platelet function in vivo, using a combination of stable biomarkers that reflect different aspects of platelet activation that go beyond the role of platelets in classical hemostasis.

Nevertheless, the present findings should be cautiously interpreted, considering several limitations. In the assessment of sex reproductive hormones, we do not measure the sex globulin binding protein that regulates the plasma levels and bioavailability of E2 and TE. We did not have an age-matched control group of males; however, as already mentioned, sex differences have already been reported in the prior platelet studies using different platelet-based tests. To explore the physiological variations of platelet biomarkers driven by endogenous sex hormones, we excluded those female subjects who were on active therapy with combined oral contraceptives (i.e., exogenous hormones), therefore limiting the generalizability of our findings to healthy premenopausal female subjects with regular cycles. Moreover, this study was unable to recruit sufficient numbers of normally ovulatory participants to solely study them, as would have been ideal. However, the appreciation of platelet marker differences between those with a late cycle progesterone value above the ovulatory threshold ( $\geq 3$ ng/ml) has yielded new insights in platelet function, particularly in Fig. 6.

From a clinical perspective, these findings raise some relevant considerations. A sex-specific approach in the evaluation of biomarkers, not limited to those related to platelet function, is very much needed, as female individuals are biologically different, and such diversity may impact atherothrombotic diseases. Over the past decade, the integration of biological sex and sociocultural gender domains in both pre-clinical and clinical research has become a priority, a suggested gateway towards equity in cardiovascular health and biomedical research[34,35]. Indeed, female subjects in reproductive age have been historically excluded from clinical studies, missing potential explanatory studies to advance our understanding of how CVD develops differentially between sexes across the lifespan. Human cells involved in atherothrombosis have a sex[36], and neglecting it will only slow down the ride towards a more effective and safe management of CVD. We envision a more sex-sensitive approach when exploring and reporting new mechanisms that implicate platelets and their heterotypic interactions across health and disease. A recently published European position statement on ischemic heart disease[37] pointed out the most updated evidence on the frequent fluctuations of prothrombotic and bleeding status related to endogenous and exogenous sex hormones (including menstrual cycle). Female representation in randomized controlled trials is mainstream, as well as considering the stage of women's lifespan to interpret findings. Finally, in the pipeline of drug development, sex and reproductive/hormonal status should be carefully assessed, especially when modulation of hemostasis is pursued[38,39].

In conclusion, we detected significantly increased levels of soluble platelet activation markers not only during menstruation, when platelets are crucial to minimize bleeding, but also in the mid-luteal phase, possibly because platelets are implicated not only in maintaining hemostasis at sites of menstrual tissue breakdown but also in modulating immune cells, maintaining vascular integrity at sites of leukocyte extravasation and in releasing factors that modulate vessel growth or that promote tissue repair. These findings support the need to consider the complexity of reproductive hormonal balance when studying platelet function, and of cautiously interpreting platelet biomarkers in clinical studies enrolling female participants in reproductive age.

## Data availability
The clinical data that support the findings of this study are available from the corresponding author upon reasonable request. All data reported in Figs. 1–6, and Supplementary Figs. S1–S4 are available in Excel file Supplementary Data 1.

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

## Acknowledgements

This study was supported by Sapienza University Starting Research Project 2013 (code: C26N13KX5R) to V.R. and co-financed by the Next Generation EU (DM 1557 11.10.2022), in the context of the National Recovery and Resilience Plan, Investment PE8—Project Age-It: "Ageing Well in an Ageing Society". The views and opinions expressed are only those of the authors and do not necessarily reflect those of the European Union or the European Commission. Neither the European Union nor the European Commission can be held responsible for them. The Visual Summary was created in BioRender. Stefanini, L. (2025) https://BioRender.com/hq0p5cc.

## Author Contributions

Conceptualization and Project administration, S.B., R.C., and L.S. Writing-Review and Editing, M.M., R.Can. and V.R.; Methodology, F.M., L.L., R.Can. and A.S.; Resources, L.N., G.F.R., T.D., A.L., C.N., A.D., G.F., R.Can. and S.S.

## Competing interests

The authors declare no competing interests

## Additional information

[1]Department of Translational and Precision Medicine, Sapienza University of Rome, Rome, Italy. [2]Department of Experimental Medicine, Sapienza University of Rome, Rome, Italy. [3]Universitat de Barcelona, Barcelona, Spain. [4]Liverpool Centre for Cardiovascular Science at University of Liverpool, Liverpool John Moores University and Liverpool Heart & Chest Hospital, Liverpool, United Kingdom. [5]Department of Clinical Internal, Anesthesiological and Cardiovascular Sciences, Sapienza University of Rome, Rome, Italy. [6]IRCCS Neuromed, 86077 Pozzilli, Italy. [7]Department of Health and Life Sciences, European University of Rome, Via degli Aldobrandeschi 190, 00163 Rome, Italy. [8]Department of Medical-Surgical Sciences and Biotechnologies, Sapienza University of Rome, Latina, Italy. [9]Italian Pasteur Institute, Cenci Bolognetti, Rome, Italy. [10]These authors contributed equally: Stefania Basili, Lucia Stefanini. ✉e-mail: marzia.miglionico@uniroma1.it

