## [Transparent Peer Review file · Communications Medicine]

Changes of in-vivo markers of platelet activation during the menstrual cycle in healthy pre-menopausal female individuals

Corresponding Author: Dr Marzia Miglionico

Version 0:

Reviewer comments:

Reviewer #1

(Remarks to the Author)

Major Comment for Authors:

This is potentially important research in studying estradiol and progesterone actions related to platelet function. Normally ovulatory menstrual cycles (all progesterone values >3 ng/ml, plus ideally a luteal length of 12 days from the urinary LH surge or 10 days by our Quantitative Basal Temperature method² mean normal estradiol values since E2 and P4 are part of a menstrual system¹. The fake diagram of menstrual cycle hormones ignores the actual levels and very real variability³ with ovulatory disturbances in 24-37% of a population-based cohort in a random cycle⁴. And a 21-35-day menstrual cycle, in women not using hormonal contraceptives, does not guarantee normal ovulation.

Specific Necessary Changes:

1. In the abstract, rather than saying menstrual cycle hormones, say estradiol and progesterone. If discussing LH and FSH describe them as gonadotrophins.
2. Remove the diagram of the menstrual cycle—it is unscientific since E2 and P4 are in different units¹ (a unitless diagram assumes all are in the same units).
3. Exclude all participants who, on day 21 did not have a progesterone value of 3 ng/ml, and whose experimental cycle length was not 21-35 days.
4. Please remove Figure 1 since it is a figment of imagination rather than reflecting physiological and scientific information.
5. Please repeat all of the important platelet analyses, PCA and all of the other data in that eligible cohort with a minimum normal menstrual cycle.
6. It would be more helpful to provide a table of the various platelet studies at baseline, what each does physiologically, the normal reference range for each and state ahead of time what would be a clinically important difference. P values show statistical and not clinical importance.
7. At the end of the introduction please state your hypothesis.
8. It would be more helpful to provide a table of the various platelet studies at baseline, what each does physiologically, the normal reference range for each and state ahead of time what would be a clinically important difference. P values show statistical and not clinical importance.

Reference List

1. Prior JC. Women's Reproductive System as Balanced Estradiol and Progesterone Actions—a revolutionary, paradigm-shifting concept in women's health. *Drug Discovery Today: Disease Models* 2020;32:31-40. doi: <https://doi.org/10.1016/j.ddmod.2020.11.005>
2. Prior JC, Vigna YM, Schulzer M, et al. Determination of luteal phase length by quantitative basal temperature methods: validation against the midcycle LH peak. *Clinical & Investigative Medicine* 1990;13:123-31.
3. Henry S, Shirin S, Goshtasebi A, et al. Prospective 1-year assessment of within-woman variability of follicular and luteal phase lengths in healthy women prescreened to have normal menstrual cycle and luteal phase lengths. *Human Reproduction* 2024;39(11):2565-74. doi: 10.1093/humrep/deae215
4. Prior JC, Naess M, Langhammer A, et al. Ovulation Prevalence in Women with Spontaneous Normal-Length Menstrual Cycles - A Population-Based Cohort from HUNT3, Norway. *PLOS One* 2015;10(8):e0134473. doi: 10.1371/journal.pone.0134473

Jerilynn C. Prior

Reviewer #2

(Remarks to the Author)

The SHOW study evaluated whether physiological fluctuations of endogenous reproductive hormones associate with changes of soluble markers of in-vivo platelet activation during physiological regular menstrual cycles in healthy pre-menopausal women aged from 18 to 40 years. Time points for blood sampling were day 1, 5, 14 and 21 of the menstrual cycle. Due to the longitudinal nature of this study, I had expect to see more advanced analysis like linear-mixed effect modeling. It is not always clear how the analysis have been conducted that takes into account that measures are taken from 1 individual. The Spearman correlation coefficient can only be used for 1 moment in time, and not for the repeated measures. Furthermore, the results are heavily focused on P-values while other metrics are more meaningful. My main suggestion would be for the authors to consult a statistician that could provide guidance on these type of analysis. See some other points of consideration below:

- Please ensure that this report adheres to a reporting guideline, for example STROBE. Please complete the STROBE reporting checklist and add this on to the supplemental file.
- Clinical data collection. More details need to be provided on definitions of variables included in this study. Have the authors used a validated questionnaire? If so, then please add these details to the manuscript. Please provide more details on how features of menstrual cycle are being collected, as for example, qualitative estimate of menstrual bleeding is very subjective without proper guidance. How is heavy menstrual bleeding defined, etc.
- The Shapiro-Wilk test is the not the best way to test for normality, suggest to use QQ plots instead.
- How are missing values handled?
- Please provide all details on the statistical analysis conducted. Also more details are needed on the regression analysis. Are variables being log transformed, etc.

Reviewer #3

(Remarks to the Author)

This manuscript presents compelling evidence that in vivo platelet activation markers vary across the menstrual cycle in healthy pre-menopausal women, offering critical insights into the dynamic interplay between reproductive hormones and hemostatic function. The study leverages the well-conceived SHOW cohort and applies robust biomarker analyses across four distinct menstrual phases. It offers novel, physiologically contextualized findings that advance our understanding of sex-specific cardiovascular biology.

The study addresses a significant knowledge gap, as women of reproductive age have historically been excluded from many cardiovascular studies, leading to a lack of precision in interpreting platelet biomarkers. The authors' emphasis on in vivo markers rather than ex vivo platelet reactivity represents a methodological and conceptual leap forward in the field.

The main findings of the study:

- The choice to use soluble platelet activation markers (TxB2, sP-selectin, sCD40L, sNOX2-dp) provides a more physiologically relevant picture than standard ex vivo aggregometry studies. The findings effectively show that platelet activation is not static but varies meaningfully with hormonal fluctuations.
- Sampling at four well-defined time points (menses, early follicular, ovulation, mid-luteal) ensures temporal granularity. This design supports the detection of dynamic, phase-specific changes, reinforcing the biological plausibility of the results.
- The application of PCA offers an elegant and unbiased validation of the primary findings. Regression models and correlation analyses further strengthen the causal inferences drawn regarding hormone-biomarker relationships.
- The study's implications for the interpretation of platelet biomarkers in both research and clinical settings are timely. As cardiovascular research increasingly moves toward sex- and gender-aware methodologies, this study provides a critical reference point.

The authors do not oversell their claims and are appropriately cautious in interpretation, especially given the small sample size.

They rightly emphasize associations over causality and acknowledge the likely indirect relationship between sex hormones and platelet activation markers.

The limitations section is transparent and comprehensive.

The manuscript is fair and balanced in its treatment of prior work.

The authors critically assess the inconsistencies in earlier studies using platelet aggregometry across the menstrual cycle. Relevant, high-quality references are cited, including both foundational and recent literature (e.g., Simon et al. on platelet transcriptomics, recent ESC reports, etc.).

The statistical analysis is sound and appropriate for the dataset.

The use of nonparametric Friedman tests for repeated measures is correct.

Multiple testing correction (Dunn-Bonferroni) was applied.

Correlations and regression models are cautiously interpreted.

PCA is well used to explore composite biomarker patterns, though additional methodological detail (see above) would enhance transparency.

Nevertheless, I have several recommendations to address:

1. While the study shows correlations between reproductive hormones and platelet biomarkers, the mechanistic pathways remain speculative. For instance, the indirect association between FSH/LH and platelet activation could be more rigorously discussed. Perhaps by referencing known systemic mediators that are hormonally regulated (e.g., endothelial function, immune cell activation, cytokines).

2. Expand on interindividual variability: the stratified analysis of sCD40L levels revealing two participant clusters is intriguing. The manuscript could benefit from deeper exploration of potential phenotypic differences (e.g., body composition, stress levels, metabolic markers) that might explain this divergence.

3. Although the authors briefly mention drug development, I recommend expanding the clinical implications section. For instance, could these findings influence how we interpret platelet activation in women presenting with acute coronary syndromes? Should menstrual phase be considered when timing diagnostic blood draws?

4. The manuscript at times conflates "sex" and "gender" (e.g., in the discussion of health equity). Given Nature's editorial standards, it is essential to clearly distinguish between these constructs throughout.

5. Given the complexity of the hormone-biomarker interactions, a graphical representation summarizing the temporal patterns and associated hormonal drivers (building on Figure 7) would enhance reader comprehension.

6. The manuscript is generally well-written, scientifically sound, and clearly organized. The language is appropriate for a specialist audience, and key findings are articulated with clarity.

Some sections, particularly the discussion, could benefit from tighter phrasing to avoid repetition.

Minor grammatical issues (e.g., "assumption of medications" instead of "use of medications") and syntax inconsistencies should be corrected for smoother readability.

More prominent summaries at the end of major sections (Results and Discussion) could help guide the reader through the main takeaways.

The manuscript is comprehensive, but slightly long for the scope of the primary findings.

The Discussion could be shortened particularly by consolidating mechanistic speculation and streamlining references to prior platelet studies.

The extended background on platelet roles outside of hemostasis is valuable, but could be more succinctly linked to the findings to maintain focus.

7. While most protocols (e.g., biomarker assays, hormonal measurements) are well described, the manuscript would benefit from:

A detailed table or summarizing assay kits used, lot numbers, sensitivity ranges, and calibrators.

More explicit detail on how PCA was conducted (e.g., software, handling of missing data).

Clarification on how "early follicular" vs. "late follicular" was operationalized given ± 2 day windows.

Minor Edits and notes

8. Line 91-92: Specify the class of estrogen/progestin used in the WHI trial to contextualize the lack of benefit.

9. Figure legends: Clearly state n-values for each figure; it is unclear if all 21 subjects had complete data across all time points.

10. Reference 6 (Simon et al.): Consider adding a sentence in the introduction to highlight platelet transcriptomic sex differences as a mechanistic basis.

Version 1:

Reviewer comments:

Reviewer #1

(Remarks to the Author)

Major Comment for Authors

Thank you for your careful attention to the suggestions I made.

Specific Necessary Changes

1. Approximately line 574—add to limitations. This study was unable to recruit sufficient numbers of normally ovulatory participants to solely study them as would have been ideal. However, the appreciation of platelet marker differences between those with a late cycle progesterone value above the ovulatory threshold (≥ 3 ng/ml) has yielded new insights in platelet function, particularly in Figure 6.

Reviewer #2

(Remarks to the Author)

The authors have engaged in the process and the manuscript is improved – this is a nice study. No further comments.

Reviewer #3

(Remarks to the Author)

I have no further comments. The authors adequately addressed my points.

Referee expertise:

Referee #1: Endocrinology and Metabolism, women's health, mensuration

Referee #2: clinical epidemiology, healthcare data and advanced statistical modelling

Referee #3: Cardio-Obstetrics, MD

Reviewers' comments:

We thank the referees for their insightful review of our manuscript and its positive evaluation. We are grateful for the changes suggested by the reviewers, which we believe have significantly improved our manuscript. Please find enclosed our response to the concerns the referees have raised.

Changes to the manuscript are underlined in the 'marked' version. Section and pages indicating changes in the manuscript are shown in this version.

Reviewer #1:

This is potentially important research in studying estradiol and progesterone actions related to platelet function. Normally ovulatory menstrual cycles (all progesterone values >3 ng/ml, plus ideally a luteal length of 12 days from the urinary LH surge or 10 days by our Quantitative Basal Temperature method² mean normal estradiol values since E2 and P4 are part of a menstrual system¹. The fake diagram of menstrual cycle hormones ignores the actual levels and very real variability with ovulatory disturbances in 24-37% of a population-based cohort in a random cycle⁴. And a 21-35-day menstrual cycle, in women not using hormonal contraceptives, does not guarantee normal ovulation.

REPLY: *We thank the reviewer for the insightful comment on how to identify normal ovulation. We incorporated this information in the interpretation of our data and made changes accordingly in the revised version of the manuscript, which we think was greatly improved by the suggested changes.*

Specific Necessary Changes:

1. In the abstract, rather than saying menstrual cycle hormones, say estradiol and progesterone. If discussing LH and FSH describe them as gonadotrophins.

REPLY: *We have amended the abstract as suggested (see page 3).*

2. Remove the diagram of the menstrual cycle—it is unscientific since E2 and P4 are in different units¹ (a unitless diagram assumes all are in the same units).

REPLY: *We agree with the reviewer and we have removed Figure 1. Instead we have generated Lowess curves that fit our data (stratified between subjects with ovulatory vs non-ovulatory cycle) and generated a summary figure 6 and revised the visual abstract.*

3. Exclude all participants who, on day 21 did not have a progesterone value of 3 ng/ml, and whose experimental cycle length was not 21-35 days.

REPLY: *We thank the reviewer for this comment. As suggested, we have considered the PG levels equal or above 3 ng/ml to identify individuals with a normal menstrual cycle. Unfortunately, the already small sample drops down to 10 individuals therefore limiting the possibility to run the same main analysis. Therefore, in the revised version of the manuscript we did not exclude subjects with PG<3 ng/ml, but we reported the differences between subjects with ovulatory versus non ovulatory cycle to reflect this important remark of the reviewer.*

4. Please remove Figure 1 since it is a figment of imagination rather than reflecting physiological and scientific information.

REPLY: *Removed as suggested.*

5. Please repeat all of the important platelet analyses, PCA and all of the other data in that eligible cohort with a minimum normal menstrual cycle.

REPLY: *As mentioned above, excluding enrolled individuals with PG<3ng/ml reduces our sample size by 50% and limits our ability to perform statistical analysis. Thus, in the revised version of the manuscript we have reported the results of the overall cohort and of the cohort stratified based on PG levels (ovulatory versus non ovulatory cycle) to reflect this important remark of the reviewer. This stratification criteria was added in the methods section (page 8, lines 158-162). We are grateful for the comment because it provides an unprecedented insight for understanding our data. For instance, in the first version of the manuscript (former Fig.4) we observed that the study participants clustered in 2 groups based on the levels of sCD40L on day 5+/-2 and we noticed that individuals with high sCD40L had the highest levels of serum progesterone, but we did not provide an explanation for this observation. Based on your comment we have now stratified the platelet parameters based on the PG levels (new Fig. 3) and we found that only study participants with an ovulatory cycle had 3-fold higher levels of sCD40L by d14. Although our data does not prove any causality and will need to be confirmed in a larger cohort, we believe this is an interesting observation that in the future could lead to a more deep understanding of the physiological role of platelets in maintaining homeostasis during the menstrual cycle.*

6. It would be more helpful to provide a table of the various platelet studies at baseline, what each does physiologically, the normal reference range for each and state ahead of time what would be a clinically important difference. P values show statistical and not clinical importance.

REPLY: *The table was added to the supplemental file (Supplementary Table 2).*

7. At the end of the introduction please state your hypothesis.

REPLY: *As suggested, we have implemented the study hypothesis at the end of the introduction (see page 6, lines 123-124).*

Reference List:

1. Prior JC. Women's Reproductive System as Balanced Estradiol and Progesterone Actions—a revolutionary, paradigm-shifting concept in women's health. *Drug Discovery Today: Disease Models* 2020;32:31-40. doi: <https://doi.org/10.1016/j.ddmod.2020.11.005>
2. Prior JC, Vigna YM, Schulzer M, et al. Determination of luteal phase length by quantitative basal temperature methods: validation against the midcycle LH peak. *Clinical & Investigative Medicine* 1990;13:123-31.
3. Henry S, Shirin S, Goshtasebi A, et al. Prospective 1-year assessment of within-woman variability of follicular and luteal phase lengths in healthy women prescreened to have normal menstrual cycle and luteal phase lengths. *Human Reproduction* 2024;39(11):2565-74. doi: 10.1093/humrep/deae215
4. Prior JC, Naess M, Langhammer A, et al. Ovulation Prevalence in Women with Spontaneous Normal-Length Menstrual Cycles - A Population-Based Cohort from HUNT3, Norway. *PLOS One* 2015;10(8):e0134473. doi: 10.1371/journal.pone.0134473

REPLY: *We thank the reviewer for these helpful references that we have added to the reference list.*

Reviewer #2:

The SHOW study evaluated whether physiological fluctuations of endogenous reproductive hormones associate with changes of soluble markers of in-vivo platelet activation during physiological regular menstrual cycles in healthy pre-menopausal women aged from 18 to 40 years. Time points for blood sampling were day 1, 5, 14 and 21 of the menstrual cycle. Due to the longitudinal nature of this study, I had expect to see more advanced analysis like linear-mixed effect modeling. It is not always clear how the analysis have been conducted that takes into account that measures are taken from 1 individual. The Spearman correlation coefficient can only be used for 1 moment in time, and not for the repeated measures. Furthermore, the results are heavily focused on P-values while other metrics are more meaningful. My main suggestion would be for the authors to consult a statistician that could provide guidance on these type of analysis.

REPLY: *We thank the reviewer for the comments. The changes among individual variables over time were analysed with a Friedman test, matching the repeated measures from the same individual. We have amended the manuscript (in the methods section, line 200, and in each figure legend) to specify that matching across different time-points was taken into account. In addition, we performed more advanced analysis as suggested. To do so, we have involved Prof Roberto Cangemi and he was added to the author list for the new analysis, the writing and the interpretation of new results. In brief, we carefully evaluated the*

*referee's comment and agreed that a more advanced type of analysis is needed to fit the structure of our data. To do so, we implemented a linear effect mixed model approach, in which we considered each platelet marker as the dependent variable; for each platelet marker, we then considered time and sexual hormones as variables with fixed effects. To account for the potential changes in the effect of sexual hormones on platelet markers, we also included in each model a time*hormone (e.g., time*FSH, time*LH) interaction term; we feel that this approach would allow for a more granular and physiological evaluation of the relationship between changes of reproductive hormones and platelet markers.*

The results of the new analyses are reported in the results section named "In vivo markers of platelet activation during the menstrual cycle" (see page 14-15).

Of note, we acknowledge the exploratory nature of our analysis, which is based on a limited sample size; for these reasons we avoided including other covariates in the models evaluated, considering the degree of freedom spent in the interaction term, and to avoid overfitting in view of our sample size.

Relevant changes on the manuscript:

Methods: Statistical Analysis (see page 9-10).

Results: We have substantially revised the Results section in accordance with the reviewers' comments and the additional statistical analyses performed. Specifically, Table 1 and several figures have been modified. The previous Figure 3 now corresponds to the current Figure 2, in which we eliminated Spearman correlation.

Discussion: We have revised the discussion section based on the suggestions received from the reviewer (page 21).

We thank the reviewer for the insightful advice which led us to significantly improve our manuscript.

See some other points of consideration below:

- Please ensure that this report adheres to a reporting guideline, for example STROBE. Please complete the STROBE reporting checklist and add this on to the supplemental file.

REPLY: *We have added the STROBE checklist as supplementary material*

- Clinical data collection. More details need to be provided on definitions of variables included in this study. Have the authors used a validated questionnaire? If so, then please add these details to the manuscript. Please provide more details on how features of menstrual cycle are being collected, as for example, qualitative estimate of menstrual bleeding is very subjective without proper guidance. How is heavy menstrual bleeding defined, etc.

REPLY: *We thank the reviewer for this comment. Overall the participants' data were self-reported and collected through an interview from clinicians as common best of practice. We have provided more details on variable definitions and clinical data collection in Methods section named Clinical Data (page 7).*

- The Shapiro-Wilk test is not the best way to test for normality, suggest to use QQ plots instead.

REPLY: We thank the reviewer for this remark. In general principles, we agree that normality tests tend to be “over-sensitive” and often give largely significant results even in case of small deviation from normality assumptions; nonetheless, this is particularly true in case of large samples (ie., due to a large power of these tests). However, we broadly agree that visual inspections of quantile-quantile (Q-Q) plots would provide a more appropriate assessment of normality distribution in the context of our study, and changed this accordingly. Of note, this approach led to superimposable interpretations and did not affect our analysis.

To reflect this change, we updated the relevant part of the methods section (see page 9, lines 198-199): “[...] Distribution of continuous variables was inspected graphically using quantile-quantile (Q-Q) plots [...]”.

- How are missing values handled?

REPLY: We thank the reviewer for this question. There were no missing data to handle in the small SHOW cohort.

- Please provide all details on the statistical analysis conducted. Also more details are needed on the regression analysis. Are variables being log transformed, etc.

REPLY: Thanks again for your comments - we have now revised our analysis in view of your previous comments and modified the statistical approach (as reported above), and detailed more in the methods section (see reply above).

Reviewer #3:

This manuscript presents compelling evidence that in vivo platelet activation markers vary across the menstrual cycle in healthy pre-menopausal women, offering critical insights into the dynamic interplay between reproductive hormones and hemostatic function. The study leverages the well-conceived SHOW cohort and applies robust biomarker analyses across four distinct menstrual phases. It offers novel, physiologically contextualized findings that advance our understanding of sex-specific cardiovascular biology.

The study addresses a significant knowledge gap, as women of reproductive age have historically been excluded from many cardiovascular studies, leading to a lack of precision in interpreting platelet biomarkers. The authors’ emphasis on in vivo markers rather than ex vivo platelet reactivity represents a methodological and conceptual leap forward in the field.

The main findings of the study:

- The choice to use soluble platelet activation markers (TxB2, sP-selectin, sCD40L, sNOX2-dp) provides a more physiologically relevant picture than standard ex vivo aggregometry studies. The findings effectively show that platelet activation is not static but varies meaningfully with hormonal fluctuations.
- Sampling at four well-defined time points (menses, early follicular, ovulation, mid-luteal) ensures temporal granularity. This design supports the detection of dynamic, phase-specific changes, reinforcing the biological plausibility of the results.
- The application of PCA offers an elegant and unbiased validation of the primary findings. Regression models and correlation analyses further strengthen the causal inferences drawn regarding hormone-biomarker relationships.
- The study's implications for the interpretation of platelet biomarkers in both research and clinical settings are timely. As cardiovascular research increasingly moves toward sex- and gender-aware methodologies, this study provides a critical reference point.

The authors do not oversell their claims and are appropriately cautious in interpretation, especially given the small sample size.

They rightly emphasize associations over causality and acknowledge the likely indirect relationship between sex hormones and platelet activation markers.

The limitations section is transparent and comprehensive.

The manuscript is fair and balanced in its treatment of prior work.

The authors critically assess the inconsistencies in earlier studies using platelet aggregometry across the menstrual cycle.

Relevant, high-quality references are cited, including both foundational and recent literature (e.g., Simon et al. on platelet transcriptomics, recent ESC reports, etc.).

The statistical analysis is sound and appropriate for the dataset.

The use of nonparametric Friedman tests for repeated measures is correct.

Multiple testing correction (Dunn-Bonferroni) was applied.

Correlations and regression models are cautiously interpreted.

PCA is well used to explore composite biomarker patterns, though additional methodological detail (see above) would enhance transparency.

REPLY: *We are glad that overall the reviewer appreciated our research project.*

Nevertheless, I have several recommendations to address:

1. While the study shows correlations between reproductive hormones and platelet biomarkers, the mechanistic pathways remain speculative. For instance, the indirect association between FSH/LH and platelet activation could be more rigorously discussed. Perhaps by referencing known systemic mediators that are hormonally regulated (e.g., endothelial function, immune cell activation, cytokines).

REPLY: *We thank the reviewer for the suggestion. While very little is known on the expression and the downstream effect of the FSH and LH receptors on immune cells, there is increasing interest in the extragonadal effect of these hormones on endothelial cells. Thus we have added the following sentence to the discussion on page 25, lines 508-511: "There is*

growing evidence that gonadotropins modulate endothelial cells, for instance by promoting the production of nitric oxide (NO), a well-established platelet inhibitor. Thus, the inverse association between FSH and LH with TxB2 and sP-selectin, could be at least in part due to the regulation of the circulating NO levels.”

2. Expand on interindividual variability: the stratified analysis of sCD40L levels revealing two participant clusters is intriguing. The manuscript could benefit from deeper exploration of potential phenotypic differences (e.g., body composition, stress levels, metabolic markers) that might explain this divergence.

REPLY: *Based on the comments of reviewer 1, in the revised version of the manuscript we have reported some of the results stratified based on PG levels, that allows to distinguish subjects who had ovulatory cycle or not according to doi: 10.1016/j.ddmod.2020.11.005. With this new approach we made the interesting observation that only study participants with an ovulatory cycle had 3-fold higher levels of sCD40L by d14, while subjects without an ovulatory cycle showed no change in the levels of sCD40L. The cluster of subjects with high sCD40L levels on day 5+/-2 and high PG levels that we have described in Fig.4 of the first version of the manuscript corresponds to the group of subjects with an ovulatory cycle. Although our data does not prove any causality and will need to be confirmed in a larger cohort, we believe that the occurrence of ovulation explains the divergence in sCD40L levels and that in the future could lead to a more deep understanding of the physiological role of platelets in maintaining homeostasis during the menstrual cycle. In the new version of the manuscript we have included a new figure (Fig.3) that shows sCD40L and the other platelet biomarkers stratified based on PG levels (ovulatory/non-ovulatory cycle) and the former Fig.4 was moved to the supplement to reduce redundancy.*

3. Although the authors briefly mention drug development, I recommend expanding the clinical implications section. For instance, could these findings influence how we interpret platelet activation in women presenting with acute coronary syndromes? Should menstrual phase be considered when timing diagnostic blood draws?

REPLY: *We thank the reviewer for pointing out the clinical implication of our findings. Currently there is a huge debate on how to take into account the sex-specific features of cardiovascular and hemostatic systems in women at risk of CVD or with CVD. A recently published position statement of EAPCI and ESC WG of Thrombosis (Paradies V, et al. Eur Heart J. 2025, doi: 10.1093/eurheartj/ehaf352, PMID: 40390370.) underlined that women experience frequent fluctuations of prothrombotic and bleeding status (related to menstrual cycle, use of oral contraceptives, hormone replacement therapy, or menopause) as well as commented on antiplatelet therapy implications in ischemic heart disease and the need of a greater female representation in randomized controlled trials. We have included this point in the discussion section (Page 26-27, lines 555-559).*

4. The manuscript at times conflates "sex" and "gender" (e.g., in the discussion of health equity). Given Nature's editorial standards, it is essential to clearly distinguish between these constructs throughout.

REPLY: *We thank the reviewer for this comment and we have amended the manuscript properly using biological sex (female vs male) and socio-cultural gender (women, men and gender-diverse people) when applicable.*

5. Given the complexity of the hormone-biomarker interactions, a graphical representation summarizing the temporal patterns and associated hormonal drivers (building on Figure 7) would enhance reader comprehension.

REPLY: *As suggested we have created a new Fig.7 in which we have graphed the data in terms of fold change relative to the first time point (day 1 of the menstrual cycle) in order to summarize the temporal changes of each measured biomarker on the same graph. Then, we have fit the data to Lowess curves to show the time course of the platelet biomarkers relative to the hormonal changes. The data was stratified based on the PG levels (PG>3ng/ml identifies ovulatory cycles) to underscore the impact of ovulation on the platelet physiological response. This new figure was also included in the graphical abstract.*

6. The manuscript is generally well-written, scientifically sound, and clearly organized. The language is appropriate for a specialist audience, and key findings are articulated with clarity. Some sections, particularly the discussion, could benefit from tighter phrasing to avoid repetition.

REPLY: *Thank you for this comment. We have checked the phrasing to avoid repetitions.*

Minor grammatical issues (e.g., "assumption of medications" instead of "use of medications") and syntax inconsistencies should be corrected for smoother readability.

REPLY: *We have checked for English grammar errors and amended.*

More prominent summaries at the end of major sections (Results and Discussion) could help guide the reader through the main takeaways.

REPLY: *We have added a take-away sentence at the end of each Result section and made a more clear summary of the major results at the beginning and at the end of the discussion.*

The manuscript is comprehensive, but slightly long for the scope of the primary findings.

REPLY: *We have shortened the introduction and the discussion and simplified the results section.*

The Discussion could be shortened particularly by consolidating mechanistic speculation and streamlining references to prior platelet studies.

REPLY: *We have shortened the overall length of the discussion, by removing the section on the meaning of the individual platelet biomarkers. Instead this information is summarized in a Table included in the supplement (Supplementary Table 2).*

The extended background on platelet roles outside of hemostasis is valuable, but could be more succinctly linked to the findings to maintain focus.

REPLY: We have removed the non-essential information from the introduction and the discussion to stay more focused.

7. While most protocols (e.g., biomarker assays, hormonal measurements) are well described, the manuscript would benefit from:

A detailed table or appendix summarizing assay kits used, lot numbers, sensitivity ranges, and calibrators.

REPLY: We thank the reviewer for the suggestion and we agree that adding more details of the kits used would be helpful especially for assays not commonly used as those for platelet biomarkers. Therefore, we included a table specifying the kit, as well as the sensitivity range, used to quantify the experimental platelet biomarkers in the supplementary materials (Supplementary Table 1).

More explicit detail on how PCA was conducted (e.g., software, handling of missing data).

REPLY: The Methods section was modified to explain how the PCA was conducted more rigorously and states as follows: The Principal component analysis (PCA) was conducted using the platelet variables (TxB2, sCD40L, sPselectin, NOX2dp) at different time points to calculate the principal components (PCs). The criteria for PCs selection were eigenvalues greater than 1. The output PC scores plot was graphed to show different symbol fill colors for different time points.

The PCA was conducted using GraphPad Prism software (Version 10.2.3) that was specified below.

Clarification on how "early follicular" vs. "late follicular" was operationalized given ± 2 day windows.

REPLY: We thank the reviewer for this comment. First day of cycle was defined by the beginning of menstruation; early follicular was defined as day 5 ± 2 to ensure capturing the last phase of menstruation, while we defined as late follicular day 7 ± 2 to ensure capturing the phase that culminates in the expected ovulation. We have clarified the timepoints in the method section.

Minor Edits and notes

8. Line 91-92: Specify the class of estrogen/progestin used in the WHI trial to contextualize the lack of benefit.

REPLY: We have specified in the revised version (conjugated equine estrogens with medroxyprogesterone acetate).

9. Figure legends: Clearly state n-values for each figure; it is unclear if all 21 subjects had complete data across all time points.

REPLY:

We apologize for the lack of clarity. In the first version of the manuscript we did not specify the n because there are no missing values in the SHOW study. However, since in the new version we present some of the data stratified by PG levels based on the comments of reviewer 1 (PG>3ng/ml are individuals with a normal ovulatory cycle), we have now included the n values in each figure legend.

10. Reference 6 (Simon et al.): Consider adding a sentence in the introduction to highlight platelet transcriptomic sex differences as a mechanistic basis.

REPLY: *We agree with the reviewer that the different transcriptomic signature could explain, at least in part, the sexual dimorphism. We included the following sentence in the introduction: The observed sexual dimorphism could be due to the different platelet transcriptomic signature between males and females (ref 6.) and/or to non-genomic effects of sex hormones directly on platelets.*

Communications Medicine is committed to improving transparency in authorship. As part of our efforts in this direction, we are now requesting that all authors identified as 'corresponding author' create and link their Open Researcher and Contributor Identifier (ORCID) with their account on the Manuscript Tracking System prior to acceptance. ORCID helps the scientific community achieve unambiguous attribution of all scholarly contributions. You can create and link your ORCID from the home page of the Manuscript Tracking System by clicking on 'Modify my Springer Nature account' and following the instructions in the link below. Please also inform all co-authors that they can add their ORCIDs to their accounts and that they must do so prior to acceptance.

REPLY: *We have added the ORCID of the corresponding author.*

Referee expertise:

Referee #1: Endocrinology and Metabolism, women's health, mensuration

Referee #2: clinical epidemiology, healthcare data and advanced statistical modelling

Referee #3: Cardio-Obstetrics, MD

We thank the referees for their insightful review of our manuscript and its positive evaluation. We are grateful for the changes suggested by the reviewers, which we believe have significantly improved our manuscript. Please find enclosed our response to the concerns the referees have raised.

Changes to the manuscript are underlined in the 'marked' version. Section and pages indicating changes in the manuscript are shown in this version.

Reviewers' comments (Revision 2):

Reviewer #1 (Remarks to the Author):

Major Comment for Authors

Thank you for your careful attention to the suggestions I made.

Specific Necessary Changes

1. Approximately line 574—add to limitations. This study was unable to recruit sufficient numbers of normally ovulatory participants to solely study them as would have been ideal. However, the appreciation of platelet marker differences between those with a late cycle progesterone value above the ovulatory threshold (≥ 3 ng/ml) has yielded new insights in platelet function, particularly in Figure 6.

REPLY: *We thank the reviewer for this comment. As suggested, we add the limitations suggested from Reviewer 1 at line 555 of underlined manuscript and at.*

Reviewer #2 (Remarks to the Author):

The authors have engaged in the process and the manuscript is improved – this is a nice study. No further comments.

REPLY: *We thank the reviewer for all the comments provided during the review process. We are pleased that, overall, he appreciated our research project.*

Reviewer #3 (Remarks to the Author):

I have no further comments. The authors adequately addressed my points.

REPLY: *We would like to thank the reviewer for all the comments that have helped to improve our manuscript.*

Reviewers' comments (Revision 1):

Reviewer #1:

This is potentially important research in studying estradiol and progesterone actions related to platelet function. Normally ovulatory menstrual cycles (all progesterone values >3 ng/ml, plus ideally a luteal length of 12 days from the urinary LH surge or 10 days by our Quantitative Basal Temperature method² mean normal estradiol values since E2 and P4 are part of a menstrual system¹. The fake diagram of menstrual cycle hormones ignores the actual levels and very real variability with ovulatory disturbances in 24-37% of a population-based cohort in a random cycle⁴. And a 21-35-day menstrual cycle, in women not using hormonal contraceptives, does not guarantee normal ovulation.

REPLY: *We thank the reviewer for the insightful comment on how to identify normal ovulation. We incorporated this information in the interpretation of our data and made changes accordingly in the revised version of the manuscript, which we think was greatly improved by the suggested changes.*

Specific Necessary Changes:

1. In the abstract, rather than saying menstrual cycle hormones, say estradiol and progesterone. If discussing LH and FSH describe them as gonadotrophins.

REPLY: *We have amended the abstract as suggested (see page 3).*

2. Remove the diagram of the menstrual cycle—it is unscientific since E2 and P4 are in different units¹ (a unitless diagram assumes all are in the same units).

REPLY: *We agree with the reviewer and we have removed Figure 1. Instead we have generated Lowess curves that fit our data (stratified between subjects with ovulatory vs non-ovulatory cycle) and generated a summary figure 6 and revised the visual abstract.*

3. Exclude all participants who, on day 21 did not have a progesterone value of 3 ng/ml, and whose experimental cycle length was not 21-35 days.

REPLY: *We thank the reviewer for this comment. As suggested, we have considered the PG levels equal or above 3 ng/ml to identify individuals with a normal menstrual cycle. Unfortunately, the already small sample drops down to 10 individuals therefore limiting the possibility to run the same main analysis. Therefore, in the revised version of the manuscript we did not exclude subjects with PG<3 ng/ml, but we reported the differences between subjects with ovulatory versus non ovulatory cycle to reflect this important remark of the reviewer.*

4. Please remove Figure 1 since it is a figment of imagination rather than reflecting physiological and scientific information.

REPLY: *Removed as suggested.*

5. Please repeat all of the important platelet analyses, PCA and all of the other data in that eligible cohort with a minimum normal menstrual cycle.

REPLY: *As mentioned above, excluding enrolled individuals with PG<3ng/ml reduces our sample size by 50% and limits our ability to perform statistical analysis. Thus, in the revised version of the manuscript we have reported the results of the overall cohort and of the cohort stratified based on PG levels (ovulatory versus non ovulatory cycle) to reflect this important remark of the reviewer. This stratification criteria was added in the methods section (page 8, lines 158-162). We are grateful for the comment because it provides an unprecedented insight for understanding our data. For instance, in the first version of the manuscript (former Fig.4) we observed that the study participants clustered in 2 groups based on the levels of sCD40L on day 5+/-2 and we noticed that individuals with high sCD40L had the highest levels of serum progesterone, but we did not provide an explanation for this observation. Based on your comment we have now stratified the platelet parameters based on the PG levels (new Fig. 3) and we found that only study participants with an ovulatory cycle had 3-fold higher levels of sCD40L by d14. Although our data does not prove any causality and will need to be confirmed in a larger cohort, we believe this is an interesting observation that in the future could lead to a more deep understanding of the physiological role of platelets in maintaining homeostasis during the menstrual cycle.*

6. It would be more helpful to provide a table of the various platelet studies at baseline, what each does physiologically, the normal reference range for each and state ahead of time what would be a clinically important difference. P values show statistical and not clinical importance.

REPLY: *The table was added to the supplemental file (Supplementary Table 2).*

7. At the end of the introduction please state your hypothesis.

REPLY: *As suggested, we have implemented the study hypothesis at the end of the introduction (see page 6, lines 123-124).*

Reference List:

1. Prior JC. Women's Reproductive System as Balanced Estradiol and Progesterone Actions—a revolutionary, paradigm-shifting concept in women's health. *Drug Discovery Today: Disease Models* 2020;32:31-40. doi: <https://doi.org/10.1016/j.ddmod.2020.11.005>
2. Prior JC, Vigna YM, Schulzer M, et al. Determination of luteal phase length by quantitative basal temperature methods: validation against the midcycle LH peak. *Clinical & Investigative Medicine* 1990;13:123-31.
3. Henry S, Shirin S, Goshtasebi A, et al. Prospective 1-year assessment of within-woman variability of follicular and luteal phase lengths in healthy women prescreened to have normal menstrual cycle and luteal phase lengths. *Human Reproduction* 2024;39(11):2565-74. doi: 10.1093/humrep/deae215
4. Prior JC, Naess M, Langhammer A, et al. Ovulation Prevalence in Women with Spontaneous Normal-Length Menstrual Cycles - A Population-Based Cohort from HUNT3, Norway. *PLOS One* 2015;10(8):e0134473. doi: 10.1371/journal.pone.0134473

REPLY: *We thank the reviewer for these helpful references that we have added to the reference list.*

Reviewer #2:

The SHOW study evaluated whether physiological fluctuations of endogenous reproductive hormones associate with changes of soluble markers of in-vivo platelet activation during physiological regular menstrual cycles in healthy pre-menopausal women aged from 18 to 40 years. Time points for blood sampling were day 1, 5, 14 and 21 of the menstrual cycle. Due to the longitudinal nature of this study, I had expect to see more advanced analysis like linear-mixed effect modeling. It is not always clear how the analysis have been conducted that takes into account that measures are taken from 1 individual. The Spearman correlation coefficient can only be used for 1 moment in time, and not for the repeated measures. Furthermore, the results are heavily focused on P-values while other metrics are more meaningful. My main suggestion would be for the authors to consult a statistician that could provide guidance on these type of analysis.

REPLY: *We thank the reviewer for the comments. The changes among individual variables over time were analysed with a Friedman test, matching the repeated measures from the same individual. We have amended the manuscript (in the methods section, line 200, and in each figure legend) to specify that matching across different time-points was taken into account. In addition, we performed more advanced analysis as suggested. To do so, we have involved Prof Roberto Cangemi and he was added to the author list for the new analysis, the writing and the interpretation of new results. In brief, we carefully evaluated the referee's comment and agreed that a more advanced type of analysis is needed to fit the structure of our data. To do so, we implemented a linear effect mixed model approach, in*

*which we considered each platelet marker as the dependent variable; for each platelet marker, we then considered time and sexual hormones as variables with fixed effects. To account for the potential changes in the effect of sexual hormones on platelet markers, we also included in each model a time*hormone (e.g., time*FSH, time*LH) interaction term; we feel that this approach would allow for a more granular and physiological evaluation of the relationship between changes of reproductive hormones and platelet markers.*

The results of the new analyses are reported in the results section named “In vivo markers of platelet activation during the menstrual cycle” (see page 14-15).

Of note, we acknowledge the exploratory nature of our analysis, which is based on a limited sample size; for these reasons we avoided including other covariates in the models evaluated, considering the degree of freedom spent in the interaction term, and to avoid overfitting in view of our sample size.

Relevant changes on the manuscript:

Methods: Statistical Analysis (see page 9-10).

Results: We have substantially revised the Results section in accordance with the reviewers' comments and the additional statistical analyses performed. Specifically, Table 1 and several figures have been modified. The previous Figure 3 now corresponds to the current Figure 2, in which we eliminated Spearman correlation.

Discussion: We have revised the discussion section based on the suggestions received from the reviewer (page 21).

We thank the reviewer for the insightful advice which led us to significantly improve our manuscript.

See some other points of consideration below:

- Please ensure that this report adheres to a reporting guideline, for example STROBE. Please complete the STROBE reporting checklist and add this on to the supplemental file.

REPLY: *We have added the STROBE checklist as supplementary material*

- Clinical data collection. More details need to be provided on definitions of variables included in this study. Have the authors used a validated questionnaire? If so, then please add these details to the manuscript. Please provide more details on how features of menstrual cycle are being collected, as for example, qualitative estimate of menstrual bleeding is very subjective without proper guidance. How is heavy menstrual bleeding defined, etc.

REPLY: *We thank the reviewer for this comment. Overall the participants' data were self-reported and collected through an interview from clinicians as common best of practice. We have provided more details on variable definitions and clinical data collection in Methods section named Clinical Data (page 7).*

- The Shapiro-Wilk test is not the best way to test for normality, suggest to use QQ plots instead.

REPLY: We thank the reviewer for this remark. In general principles, we agree that normality tests tend to be “over-sensitive” and often give largely significant results even in case of small deviation from normality assumptions; nonetheless, this is particularly true in case of large samples (ie., due to a large power of these tests). However, we broadly agree that visual inspections of quantile-quantile (Q-Q) plots would provide a more appropriate assessment of normality distribution in the context of our study, and changed this accordingly. Of note, this approach led to superimposable interpretations and did not affect our analysis.

To reflect this change, we updated the relevant part of the methods section (see page 9, lines 198-199): “[...] Distribution of continuous variables was inspected graphically using quantile-quantile (Q-Q) plots [...]”.

- How are missing values handled?

REPLY: We thank the reviewer for this question. There were no missing data to handle in the small SHOW cohort.

- Please provide all details on the statistical analysis conducted. Also more details are needed on the regression analysis. Are variables being log transformed, etc.

REPLY: Thanks again for your comments - we have now revised our analysis in view of your previous comments and modified the statistical approach (as reported above), and detailed more in the methods section (see reply above).

Reviewer #3:

This manuscript presents compelling evidence that in vivo platelet activation markers vary across the menstrual cycle in healthy pre-menopausal women, offering critical insights into the dynamic interplay between reproductive hormones and hemostatic function. The study leverages the well-conceived SHOW cohort and applies robust biomarker analyses across four distinct menstrual phases. It offers novel, physiologically contextualized findings that advance our understanding of sex-specific cardiovascular biology.

The study addresses a significant knowledge gap, as women of reproductive age have historically been excluded from many cardiovascular studies, leading to a lack of precision in interpreting platelet biomarkers. The authors’ emphasis on in vivo markers rather than ex vivo platelet reactivity represents a methodological and conceptual leap forward in the field.

The main findings of the study:

- The choice to use soluble platelet activation markers (TxB2, sP-selectin, sCD40L, sNOX2-dp) provides a more physiologically relevant picture than standard ex vivo aggregometry studies. The findings effectively show that platelet activation is not static but varies meaningfully with hormonal fluctuations.
- Sampling at four well-defined time points (menses, early follicular, ovulation, mid-luteal) ensures temporal granularity. This design supports the detection of dynamic, phase-specific changes, reinforcing the biological plausibility of the results.
- The application of PCA offers an elegant and unbiased validation of the primary findings. Regression models and correlation analyses further strengthen the causal inferences drawn regarding hormone-biomarker relationships.
- The study's implications for the interpretation of platelet biomarkers in both research and clinical settings are timely. As cardiovascular research increasingly moves toward sex- and gender-aware methodologies, this study provides a critical reference point.

The authors do not oversell their claims and are appropriately cautious in interpretation, especially given the small sample size.

They rightly emphasize associations over causality and acknowledge the likely indirect relationship between sex hormones and platelet activation markers.

The limitations section is transparent and comprehensive.

The manuscript is fair and balanced in its treatment of prior work.

The authors critically assess the inconsistencies in earlier studies using platelet aggregometry across the menstrual cycle.

Relevant, high-quality references are cited, including both foundational and recent literature (e.g., Simon et al. on platelet transcriptomics, recent ESC reports, etc.).

The statistical analysis is sound and appropriate for the dataset.

The use of nonparametric Friedman tests for repeated measures is correct.

Multiple testing correction (Dunn-Bonferroni) was applied.

Correlations and regression models are cautiously interpreted.

PCA is well used to explore composite biomarker patterns, though additional methodological detail (see above) would enhance transparency.

REPLY: *We are glad that overall the reviewer appreciated our research project.*

Nevertheless, I have several recommendations to address:

1. While the study shows correlations between reproductive hormones and platelet biomarkers, the mechanistic pathways remain speculative. For instance, the indirect association between FSH/LH and platelet activation could be more rigorously discussed. Perhaps by referencing known systemic mediators that are hormonally regulated (e.g., endothelial function, immune cell activation, cytokines).

REPLY: *We thank the reviewer for the suggestion. While very little is known on the expression and the downstream effect of the FSH and LH receptors on immune cells, there is increasing interest in the extragonadal effect of these hormones on endothelial cells. Thus we have added the following sentence to the discussion on page 25, lines 508-511: "There is*

growing evidence that gonadotropins modulate endothelial cells, for instance by promoting the production of nitric oxide (NO), a well-established platelet inhibitor. Thus, the inverse association between FSH and LH with TxB2 and sP-selectin, could be at least in part due to the regulation of the circulating NO levels.”

2. Expand on interindividual variability: the stratified analysis of sCD40L levels revealing two participant clusters is intriguing. The manuscript could benefit from deeper exploration of potential phenotypic differences (e.g., body composition, stress levels, metabolic markers) that might explain this divergence.

REPLY: *Based on the comments of reviewer 1, in the revised version of the manuscript we have reported some of the results stratified based on PG levels, that allows to distinguish subjects who had ovulatory cycle or not according to doi: 10.1016/j.ddmod.2020.11.005. With this new approach we made the interesting observation that only study participants with an ovulatory cycle had 3-fold higher levels of sCD40L by d14, while subjects without an ovulatory cycle showed no change in the levels of sCD40L. The cluster of subjects with high sCD40L levels on day 5+/-2 and high PG levels that we have described in Fig.4 of the first version of the manuscript corresponds to the group of subjects with an ovulatory cycle. Although our data does not prove any causality and will need to be confirmed in a larger cohort, we believe that the occurrence of ovulation explains the divergence in sCD40L levels and that in the future could lead to a more deep understanding of the physiological role of platelets in maintaining homeostasis during the menstrual cycle. In the new version of the manuscript we have included a new figure (Fig.3) that shows sCD40L and the other platelet biomarkers stratified based on PG levels (ovulatory/non-ovulatory cycle) and the former Fig.4 was moved to the supplement to reduce redundancy.*

3. Although the authors briefly mention drug development, I recommend expanding the clinical implications section. For instance, could these findings influence how we interpret platelet activation in women presenting with acute coronary syndromes? Should menstrual phase be considered when timing diagnostic blood draws?

REPLY: *We thank the reviewer for pointing out the clinical implication of our findings. Currently there is a huge debate on how to take into account the sex-specific features of cardiovascular and hemostatic systems in women at risk of CVD or with CVD. A recently published position statement of EAPCI and ESC WG of Thrombosis (Paradies V, et al. Eur Heart J. 2025, doi: 10.1093/eurheartj/ehaf352, PMID: 40390370.) underlined that women experience frequent fluctuations of prothrombotic and bleeding status (related to menstrual cycle, use of oral contraceptives, hormone replacement therapy, or menopause) as well as commented on antiplatelet therapy implications in ischemic heart disease and the need of a greater female representation in randomized controlled trials. We have included this point in the discussion section (Page 26-27, lines 555-559).*

4. The manuscript at times conflates "sex" and "gender" (e.g., in the discussion of health equity). Given Nature's editorial standards, it is essential to clearly distinguish between these constructs throughout.

REPLY: *We thank the reviewer for this comment and we have amended the manuscript properly using biological sex (female vs male) and socio-cultural gender (women, men and gender-diverse people) when applicable.*

5. Given the complexity of the hormone-biomarker interactions, a graphical representation summarizing the temporal patterns and associated hormonal drivers (building on Figure 7) would enhance reader comprehension.

REPLY: *As suggested we have created a new Fig.7 in which we have graphed the data in terms of fold change relative to the first time point (day 1 of the menstrual cycle) in order to summarize the temporal changes of each measured biomarker on the same graph. Then, we have fit the data to Lowess curves to show the time course of the platelet biomarkers relative to the hormonal changes. The data was stratified based on the PG levels (PG>3ng/ml identifies ovulatory cycles) to underscore the impact of ovulation on the platelet physiological response. This new figure was also included in the graphical abstract.*

6. The manuscript is generally well-written, scientifically sound, and clearly organized. The language is appropriate for a specialist audience, and key findings are articulated with clarity. Some sections, particularly the discussion, could benefit from tighter phrasing to avoid repetition.

REPLY: *Thank you for this comment. We have checked the phrasing to avoid repetitions.*

Minor grammatical issues (e.g., "assumption of medications" instead of "use of medications") and syntax inconsistencies should be corrected for smoother readability.

REPLY: *We have checked for English grammar errors and amended.*

More prominent summaries at the end of major sections (Results and Discussion) could help guide the reader through the main takeaways.

REPLY: *We have added a take-away sentence at the end of each Result section and made a more clear summary of the major results at the beginning and at the end of the discussion.*

The manuscript is comprehensive, but slightly long for the scope of the primary findings.

REPLY: *We have shortened the introduction and the discussion and simplified the results section.*

The Discussion could be shortened particularly by consolidating mechanistic speculation and streamlining references to prior platelet studies.

REPLY: *We have shortened the overall length of the discussion, by removing the section on the meaning of the individual platelet biomarkers. Instead this information is summarized in a Table included in the supplement (Supplementary Table 2).*

The extended background on platelet roles outside of hemostasis is valuable, but could be more succinctly linked to the findings to maintain focus.

REPLY: We have removed the non-essential information from the introduction and the discussion to stay more focused.

7. While most protocols (e.g., biomarker assays, hormonal measurements) are well described, the manuscript would benefit from:

A detailed table or appendix summarizing assay kits used, lot numbers, sensitivity ranges, and calibrators.

REPLY: We thank the reviewer for the suggestion and we agree that adding more details of the kits used would be helpful especially for assays not commonly used as those for platelet biomarkers. Therefore, we included a table specifying the kit, as well as the sensitivity range, used to quantify the experimental platelet biomarkers in the supplementary materials (Supplementary Table 1).

More explicit detail on how PCA was conducted (e.g., software, handling of missing data).

REPLY: The Methods section was modified to explain how the PCA was conducted more rigorously and states as follows: The Principal component analysis (PCA) was conducted using the platelet variables (TxB2, sCD40L, sPselectin, NOX2dp) at different time points to calculate the principal components (PCs). The criteria for PCs selection were eigenvalues greater than 1. The output PC scores plot was graphed to show different symbol fill colors for different time points.

The PCA was conducted using GraphPad Prism software (Version 10.2.3) that was specified below.

Clarification on how "early follicular" vs. "late follicular" was operationalized given ± 2 day windows.

REPLY: We thank the reviewer for this comment. First day of cycle was defined by the beginning of menstruation; early follicular was defined as day 5 ± 2 to ensure capturing the last phase of menstruation, while we defined as late follicular day 7 ± 2 to ensure capturing the phase that culminates in the expected ovulation. We have clarified the timepoints in the method section.

Minor Edits and notes

8. Line 91-92: Specify the class of estrogen/progestin used in the WHI trial to contextualize the lack of benefit.

REPLY: We have specified in the revised version (conjugated equine estrogens with medroxyprogesterone acetate).

9. Figure legends: Clearly state n-values for each figure; it is unclear if all 21 subjects had complete data across all time points.

REPLY:

We apologize for the lack of clarity. In the first version of the manuscript we did not specify the n because there are no missing values in the SHOW study. However, since in the new version we present some of the data stratified by PG levels based on the comments of reviewer 1 (PG>3ng/ml are individuals with a normal ovulatory cycle), we have now included the n values in each figure legend.

10. Reference 6 (Simon et al.): Consider adding a sentence in the introduction to highlight platelet transcriptomic sex differences as a mechanistic basis.

REPLY: *We agree with the reviewer that the different transcriptomic signature could explain, at least in part, the sexual dimorphism. We included the following sentence in the introduction: The observed sexual dimorphism could be due to the different platelet transcriptomic signature between males and females (ref 6.) and/or to non-genomic effects of sex hormones directly on platelets.*

Communications Medicine is committed to improving transparency in authorship. As part of our efforts in this direction, we are now requesting that all authors identified as 'corresponding author' create and link their Open Researcher and Contributor Identifier (ORCID) with their account on the Manuscript Tracking System prior to acceptance. ORCID helps the scientific community achieve unambiguous attribution of all scholarly contributions. You can create and link your ORCID from the home page of the Manuscript Tracking System by clicking on 'Modify my Springer Nature account' and following the instructions in the link below. Please also inform all co-authors that they can add their ORCIDs to their accounts and that they must do so prior to acceptance.

REPLY: *We have added the ORCID of the corresponding author.*